# Atomic order of rare earth ions in a complex oxide: a path to magnetotaxial anisotropy

**Allison C. Kaczmarek** [1,4] ✉, **Ethan R. Rosenberg**[1,2,4], **Yixuan Song** [1], **Kevin Ye**[1], **Gavin A. Winter**[1], **Aubrey N. Penn**[3], **Rafael Gomez-Bombarelli** [1], **Geoffrey S. D. Beach** [1] **& Caroline A. Ross** [1]

Complex oxides offer rich magnetic and electronic behavior intimately tied to the composition and arrangement of cations within the structure. Rare earth iron garnet films exhibit an anisotropy along the growth direction which has long been theorized to originate from the ordering of different cations on the same crystallographic site. Here, we directly demonstrate the three-dimensional ordering of rare earth ions in pulsed laser deposited $(Eu_xTm_{1-x})_3Fe_5O_{12}$ garnet thin films using both atomically-resolved elemental mapping to visualize cation ordering and X-ray diffraction to detect the resulting order superlattice reflection. We quantify the resulting ordering-induced 'magnetotaxial' anisotropy as a function of Eu:Tm ratio using transport measurements, showing an overwhelmingly dominant contribution from magnetotaxial anisotropy that reaches 30 kJ m$^{-3}$ for garnets with x = 0.5. Control of cation ordering on inequivalent sites provides a strategy to control matter on the atomic level and to engineer the magnetic properties of complex oxides.

Neumann's principle states that the symmetry elements of any physical property must include the symmetry elements of the point group of the crystal[1]. When equivalent sites in a crystal are occupied by more than one atomic species, site ordering of the atoms lowers the symmetry[2] and dramatically affects the crystal properties, including magnetic and electronic behavior[3]. The perovskite family ($ABO_3$) provides rich examples of antiferro-, ferri-, ferromagnetic or noncollinear magnetic order promoted by ordering of mixed $B$-site cations[4–7], in contrast to the spin-glass behavior typical of the disordered perovskite. Spontaneous $B$-site ordering is driven by differences in ionic charge or radius[5], but site ordering can also be introduced artificially using atomic scale layer-by-layer growth[8], as in the (111)-ordered double perovskites $Sr_2FeRuO_6$ [9] and $La_2CrFeO_6$[10] (both room-temperature ferromagnets) and multiferroic $Bi_2FeCoO_6$[11]. Expanding site-ordering strategies to obtain a specific 3D ordering of atoms could produce new materials with unprecedented properties, beyond what can be achieved with synthetic layer-by-layer growth.

Rare earth iron garnets (REIG, $RE_3Fe_2Fe_3O_{12}$) have cubic Ia$\bar{3}$d symmetry with each formula unit containing 3 $RE^{3+}$ ions on the dodecahedral $c$ sites, 2 $Fe^{3+}$ ions on the octahedral $a$ sites, and 3 $Fe^{3+}$ ions on the tetrahedral $d$ sites, shown in Fig. 1a. Iron garnets are ferrimagnetic: $a$ and $d$ site $Fe^{3+}$ exhibit antiparallel coupling by superexchange through the oxygen ligands, and the $c$ site cation, if magnetic, couples antiparallel to the $d$ site $Fe^{3+}$. Iron garnet thin films have been developed for magnetic bubble memory[12], integrated magnetooptical devices[13], and spintronic logic and memory applications[14–17], many of which require films with perpendicular magnetic anisotropy (PMA).

In films of iron garnets with mixed RE ions, Callen proposed[18] that ordering of the cations on the dodecahedral sites lowers the symmetry and introduces a source of magnetic anisotropy additional to the contributions[19] from shape (magnetostatic anisotropy, $K_{MS}$), crystal symmetry (magnetocrystalline anisotropy, $K_{MC}$), and strain (magnetoelastic anisotropy, $K_{ME}$). This ordering-induced

[1]Department of Materials Science and Engineering, Massachusetts Institute of Technology, Cambridge, MA 02139, USA. [2]Lawrence Livermore National Laboratory, Livermore, CA 94550, USA. [3]MIT.nano, Massachusetts Institute of Technology, Cambridge, MA 02139, USA. [4]These authors contributed equally: Allison C. Kaczmarek, Ethan R. Rosenberg. ✉e-mail: kacz@mit.edu

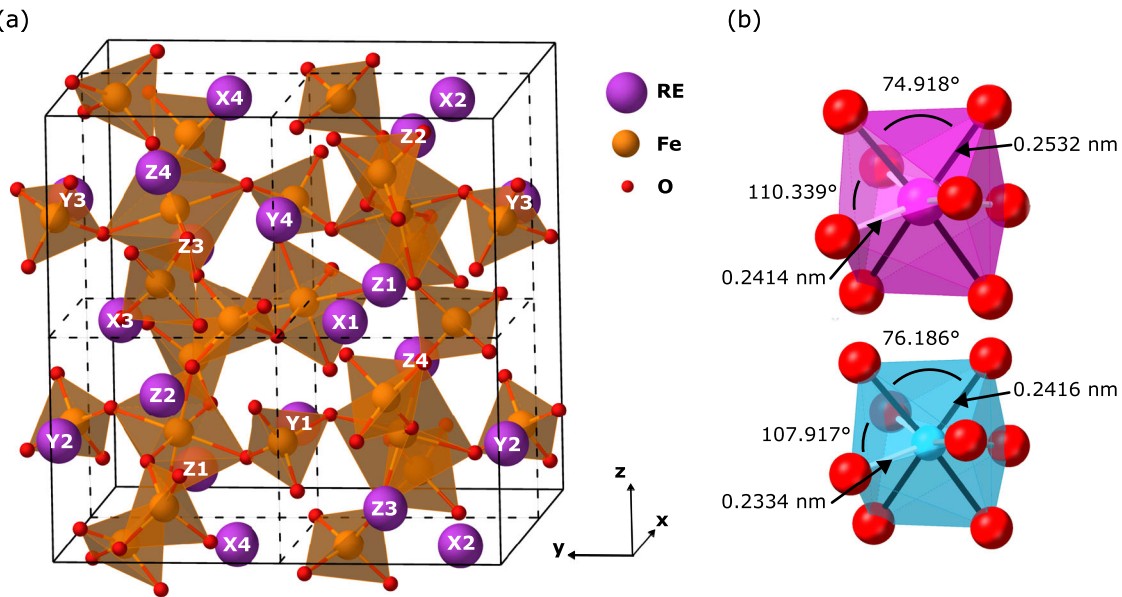

**Fig. 1 | Garnet crystal structure. a** Half of the garnet unit cell, showing the labels (e.g. X1, Y2, Z3) and locations of rare earth ions within the unit cell. **b** Distorted dodecahedra in the mixed EuTmIG structure for larger (pink, $Eu^{3+}$) and smaller (blue, $Tm^{3+}$) rare earth ions.

anisotropy was described by a site-preference model (Supplementary Note 1) that accounts for preferential site occupancy of non-equivalent dodecahedral sites on the growth surface. For example, in the (111) plane, the 24 $c$ sites in the unit cell fall into two sets, inequivalent due to their different orientations with respect to the [111] direction. Filling these $c$ sites with ions of different sizes distorts polyhedral geometries (Fig. 1b), affecting the magnetic exchange and total anisotropy. Therefore, during the growth of (111) iron garnet films, segregation of RE ions between the inequivalent $c$ sites due to steric differences can lead to ordering within the unit cell and hence yield an additional anisotropy term. Other orientations such as (110), (112) or (001) correspond to different sets of symmetrically inequivalent $c$ sites leading to anisotropies with different magnitudes and symmetries. Callen's model does not explain how this site order would manifest across many unit cells of the material when more than one order variant is possible.

This anisotropy term found in films of mixed garnets has been known as "growth-induced" anisotropy, but here we will refer to it as "magnetotaxial" anisotropy, a broader term which emphasizes the origin of the anisotropy rather than the mechanism by which it is achieved. Specifically, the choice of the root *taxis*, the Greek word for battle array, order, or regularity, highlights the importance of arrangement over growth kinetics, so that in future investigations, anisotropy arising generally from artificial site ordering could also be described as magnetotaxial. This nomenclature also motivates the investigation of anisotropies with other symmetries that arise from ionic or defect site ordering. For a mixture of two $c$-site ions, the site-preference model predicts a contribution to anisotropy approaching the quadratic form:

$$K_{MT} = K(x)(1 - x), \qquad (1)$$

where $x$ is the atomic fraction of one of the RE ions. The coefficient $K$ increases with the ionic radius difference between the two ions, reflecting the role of site size and steric effects[20,21]. The site ordering is metastable and vanishes if the garnet film is annealed at high temperatures, as has been shown for liquid phase-grown garnets annealed at 1250 °C for tens of hours[22]. Additional magnetotaxial anisotropy contributions can arise from the site ordering of atoms and vacancies on the $a$ and $d$ sites of the garnet structure[23–26].

While growth-induced anisotropy has been described and parameterized in REIGs grown by liquid and vapor phase epitaxy, no direct proof of site ordering or understanding of the uniaxial anisotropy arising from lowered symmetry in these materials exists. Here, we demonstrate through both X-ray diffraction and direct elementally resolved mapping that magnetotaxial anisotropy stems from a strong site preference in a mixed REIG, leading to a three-dimensionally ordered RE sublattice. Furthermore, we provide mechanistic insight as to how the ensemble of ordered unit cells produces a net uniaxial anisotropy out of the film plane. We quantify the emergent anisotropy and show using first principles modeling that site ordering leads to symmetry reduction and magnetotaxial anisotropy. These results resolve longstanding questions concerning the origins of anisotropy in garnet materials and demonstrate the power of atomic ordering to engineer materials with unique magnetic properties.

## Results

To identify the magnetotaxial anisotropy, we selected a mixed REIG and a substrate combination for which strain, and thereby magnetoelastic anisotropy, is minimized when the magnetotaxial anisotropy is expected to be largest. Eu and Tm were chosen as the RE ions, and (111) gadolinium gallium garnet ($Gd_3Ga_5O_{12}$, GGG) was chosen as the substrate. The lattice parameter of EuIG is smaller than that of GGG, and its magnetostriction, $\lambda_{111}$, is negative. Meanwhile, TmIG has a larger lattice parameter than GGG, and its $\lambda_{111}$ is positive[12]. In each case, the magnetoelastic anisotropy overcomes the shape anisotropy, and the end-member films exhibit PMA. However, at intermediate compositions, the lattice parameter of $(Eu_xTm_{1-x})_3IG$ matches that of GGG, and $\lambda_{111}$ passes through zero when the fraction of EuIG, $x$, is 0.8. In fact, there is a range of compositions for which $\lambda_{111}$ is negative, the film is in a state of in-plane compression, and the magnetoelastic anisotropy contributes to an in-plane easy axis of magnetization. Thus, for intermediate compositions the magnetoelastic anisotropy is small, and any magnetotaxial anisotropy is clearly revealed.

A series of $(Eu_xTm_{1-x})_3IG$ films with a thickness of 16–40 nm were grown on (111) GGG by pulsed laser codeposition[27]. By fitting the high-resolution X-ray diffraction (HRXRD) symmetric scans about the (444) reflection (Fig. 2a, Supplementary Note 2), we determined the

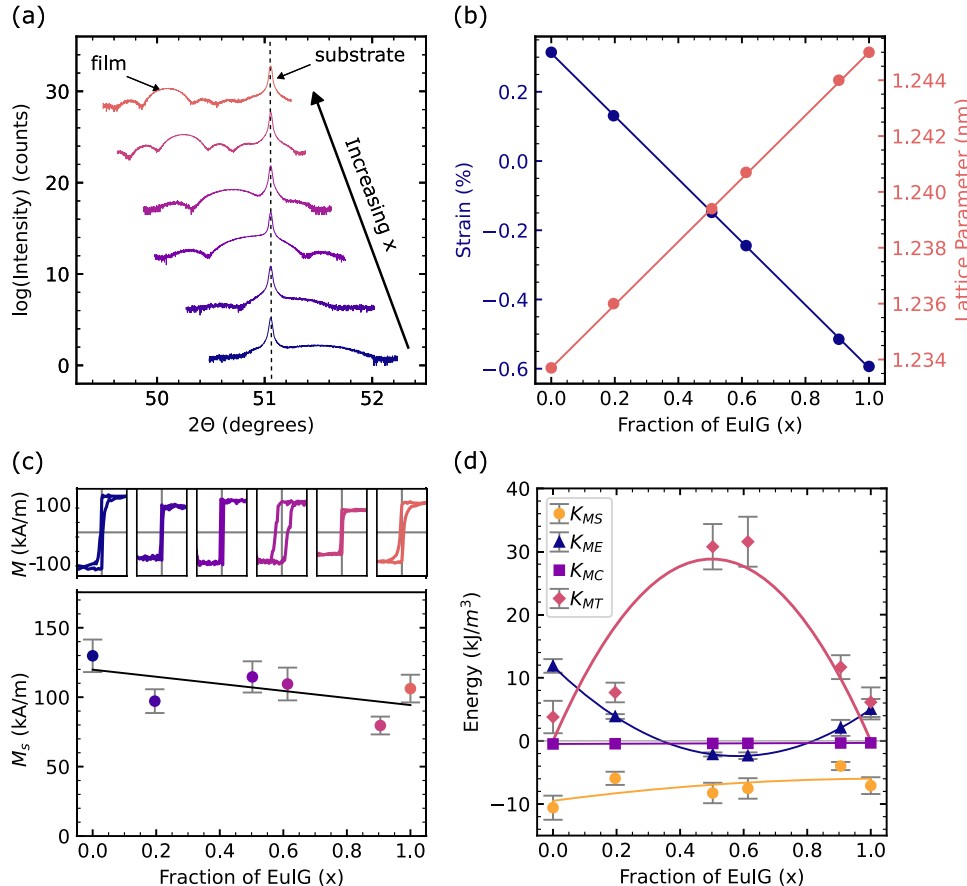

**Fig. 2 | Magnetotaxial anisotropy in a composition series of EuTmIG films.**
**a** Symmetric 2θ diffraction scans of the composition series films about the (444)
reflection. **b** Strain and lattice parameter of each film extracted from HRXRD as a
function of Eu content in the mixed garnet films. **c** Out-of-plane magnetic

hysteresis loops and saturation magnetization as a function of Eu content in the
mixed garnet films. The field range for each inset loop is ±100 A/m.
**d** Contributions to magnetic anisotropy extracted from SMR measurements. In
each graph, error bars represent one standard deviation.

thickness and out-of-plane lattice parameter for each film, from which
we calculate the strain and the Eu:Tm ratio. The lattice parameters of
mixed REIGs vary nearly linearly between the end members, consistent
with Vegard's law. Figure 2b shows that as $x$ increases, the shear strain
(i.e. the deviation of the unit cell corner angle, $\beta$, from the unstrained
value of 90°) changes linearly, and at a composition around $x = 0.35$
the strain passes through zero.

Magnetic hysteresis loops, obtained by vibrating sample magne-
tometry (VSM), are square in the out-of-plane direction, showing that
the films have PMA across the entire composition range (Fig. 2c,
Supplementary Note 3), even when the magnetoelastic anisotropy is
small. The saturation magnetization $M_s$ exhibits a slight decrease with
increasing $x$ indicating $Eu^{3+}$ in garnets contributes a larger moment
than $Tm^{3+}$ at room temperature.

The net magnetic anisotropy energy (MAE), $K_{U,eff}$ in $(Eu_xTm_{1-x})_3IG$
films was quantified by spin-Hall magnetoresistance (SMR) measure-
ments (Supplementary Note 4), where positive MAE corresponds to
PMA. The contributions to total anisotropy from magnetoelastic,
magnetostatic, and magnetocrystalline anisotropy are given by the
following equation:[28,29]

$$K_{U,eff} = K_{ME} + K_{MS} + K_{MC} + K_{MT} \qquad (2)$$

$$K_{U,eff} = \frac{9}{4}\lambda_{111}c_{44}\left(\frac{\pi}{2} - \beta\right) - \left(\frac{\mu_0}{2}\right)M_s^2 + \frac{K_1}{12} + K_{MT} \qquad (3)$$

Based on the measured shear strain $\frac{\pi}{2} - \beta$ (Supplementary Note 2)
and saturation magnetization $M_s$, and taking a linear interpolation of $K_1$

between that of the end-members, we determine $K_{ME}$, $K_{MS}$, $K_{MC}$ and
hence the magnetotaxial anisotropy, $K_{MT}$, Fig. 2d, Supplementary
Table 4. $K_{MT}$ is quadratic in $x$ as expected for growth-induced aniso-
tropy, but with magnitude up to 30 kJ m$^{-3}$, considerably greater than
typical values for iron garnet films grown by LPE[12]. Moreover, $K_{MT}$ is the
dominant anisotropy term over a wide range of composition, except
for compositions near the end-members ($x = 0$ and $x = 1$).

Having established the presence of a large magnetotaxial aniso-
tropy, we then demonstrated its origins in the three-dimensional site
ordering of the Tm and Eu ions, first on a global scale by diffraction and
then on a local scale by elementally-resolved electron microscopy.
Ordering of ions within the unit cell leads to non-zero structure factors
and the appearance of additional diffraction peaks. We emphasize that
the RE ordering is not a simple arrangement of columns of each type of
ion along the growth direction, nor layers of each type of cation par-
allel to the growth plane. For example, along the [111] growth direction,
the ordering yields alternating double layers and mixed layers of ions
along (110) (Fig. S1). The unit cell is reduced from body-centered cubic
to trigonal which unlocks the forbidden ($\bar{1}01$) reflection at $2\theta \cong 12°$ (for
Co Kα radiation with wavelength 1.79 Å; the exact peak position varies
with RE composition).

Figure 3a shows the presence of the ($\bar{1}01$), ($01\bar{1}$) and ($1\bar{1}0$) reflec-
tions in the EuTmIG film with $x \cong 0.5$. The presence of three peaks
arises from the three possible orientational variants of the 3D cation-
ordering visualized in Fig. 3b, and they exist in roughly equivalent
proportions, as evidenced by peak intensity. Exchange averaging of
the locally varying uniaxial anisotropy from the three orientational
variants produces a net uniaxial anisotropy with its axis along [111]

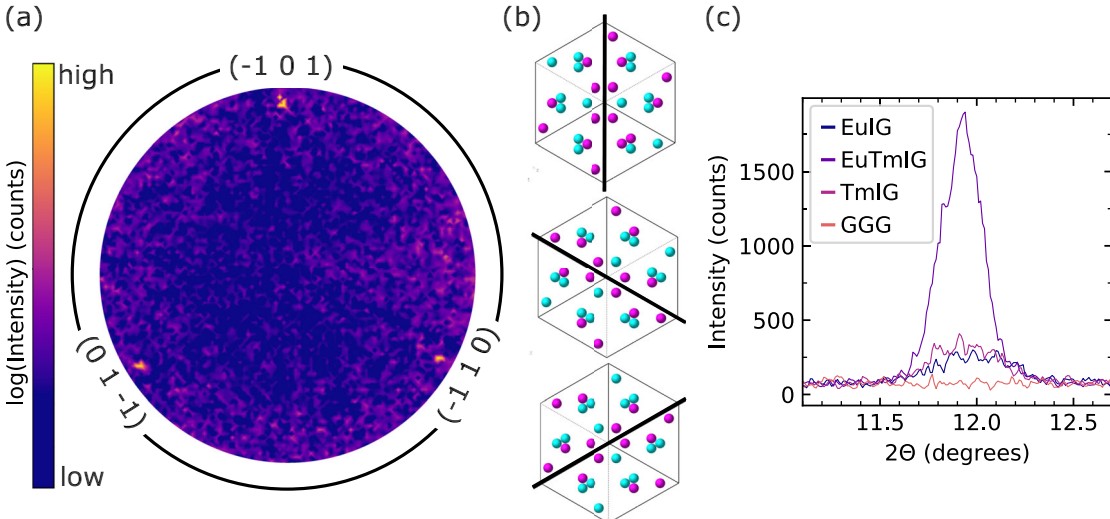

**Fig. 3 | Proof of superlattice peaks in ordered EuTmIG. a** Pole figure (phi/chi scan) of EuTmIG ($x = 0.5$) shows {110} peaks, indicative of RE order within the unit cell. **b** (111) projection of the three ways RE ions can order within the garnet unit cell. **c** 2θ scans of the ($\bar{1}$01) peak in three REIG thin films and the GGG substrate.

(a discussion of this result is found in Supplementary Note 5). Figure 3c compares the superlattice-like order reflection with those of the end-member EuIG and TmIG films and a bare GGG substrate. For the GGG substrate, the peak is absent as expected since all $c$ sites are occupied by $Gd^{3+}$. For the EuIG and TmIG this peak is present but small, suggesting that there is some ordering on the dodecahedral sites, perhaps due to vacancies or antisite defects produced during growth. Supplementary Note 8 shows via modeling that the weak intensity for the end member films can arise from point defects such as vacancies. Furthermore, the atomic number difference between Eu and Tm in the ordered structure is sufficient to yield a superlattice peak whose intensity depends on composition, the degree of site ordering, and growth orientation.

Next, we visualize the local ordering of RE ions directly using atomically resolved elemental mapping. Callen's model predicts that site ordering will be present for (111), (110) and other growth surfaces (Supplementary Note 1), but to detect the ordering in TEM it is necessary for there to be a zone axis parallel to columns of dissimilar RE ions. We therefore selected a (110)-oriented $(Eu_xTm_{1-x})_3IG$ film with $x = 0.5$ because the lower symmetry allows us to identify a zone axis parallel to columns of inequivalent $c$ sites (labeled α, β, γ in Fig. 4c), and to image the atomic order along this zone axis using energy dispersive spectroscopy (EDS) elemental mapping. This analysis cannot be done for the (111)-oriented films due to the existence of three variants of the site-ordered structure; superposition of the variants averages the atomic ordering along any zone axis and precludes visualization of the ordering (Supplementary Note 6). Considering the non-degenerate orientations of the $c$ sites at the (110) surface, the site-preference model predicts that the three distinct $c$ site groups will be visible in columns in the [$1\bar{1}1$] projection, which lies within the film plane (Supplementary Note 1). Along this zone axis the columns of inequivalent $c$ sites appear as groups of three and rings of six in Fig. 4c.

Figure 4a shows the scanning transmission electron microscopy (STEM) high angle annular dark field (HAADF) image and Fig. 4b the EDS elemental maps of Eu, Tm, and Fe along the [$1\bar{1}1$] zone axis of the (110) film, derived from Eu, Tm, and Fe X-ray peaks with minimal overlap in their energy range (Supplementary Note 6). The EDS (Fig. 4c, d) shows that Eu prefers α sites while Tm prefers β and γ sites, but is also present in α sites. Since there are twice as many α sites as β and γ combined, it is unavoidable that some Tm must occupy the α

sites considering the overall composition of $x = 0.5$. We also identified ordered antisite defects of $RE_{Fe}$ and $Fe_{RE}$ (Supplementary Note 6). The EDS analysis therefore confirms the ordering of Eu and Tm in the garnet, and gives direct evidence of the atomic arrangement responsible for the measured magnetotaxial anisotropy. Supplementary Note 9 describes XRD of the (110)-oriented films, confirming the presence of RE site ordering.

Finally, we consider the effect of RE ordering on the structure and magnetic anisotropy using density functional theory (DFT; Supplementary Note 7). For a (111)-oriented film, ordering of Eu and Tm in the inequivalent sites of Fig. 1a leads to a symmetry reduction of the unit cell as well as local distortions of the $c$ site coordination polyhedra, Fig. 1b. The structural changes modify the orbital overlap between neighbors, thus changing the magnetic exchange and producing magnetic anisotropy. The symmetry of the oxygen coordination around the Fe cations is also reduced by $c$ site ordering, which likely explains the PMA observed in iron garnets with mixed non-magnetic $c$ site ions, such as $Bi^{3+}$ and $Y^{3+}$ in BiYIG, for which the ordered species neither need to be magnetic or a rare earth element to induce substantial anisotropy[30]. For the EuTmIG DFT model, the magnetic anisotropy was calculated as the energy difference between magnetization along the [111] (out-of-plane) and [$\bar{1}$01] (in plane) directions, which includes zero temperature magnetocrystalline and magnetotaxial anisotropy contributions. This anisotropy was about two times larger for the ordered EuTmIG (140.6 kJ m$^{-3}$) compared to the EuIG (75.8 kJ m$^{-3}$) and TmIG (−46.6 kJ m$^{-3}$) end members, supporting the hypothesis that $c$ site ordering is responsible for an important contribution to anisotropy which drives PMA in mixed REIG films.

## Discussion

Our results unambiguously demonstrate the existence of dodecahedral cation site ordering of RE ions in iron garnet films based on both X-ray diffraction and elemental mapping. Rare earth ordering was proposed five decades ago by Callen[18] to explain the perpendicular magnetic anisotropy of iron garnet films grown by liquid phase epitaxy, but no prior measurements had, until now, definitively proved its origin in the site ordering of rare earth cations. Site ordering originates from the non-degeneracy of the dodecahedral sites on the growth surface and yields a 3D cation order, and an associated magnetotaxial anisotropy, that is specific to the growth direction of the film.

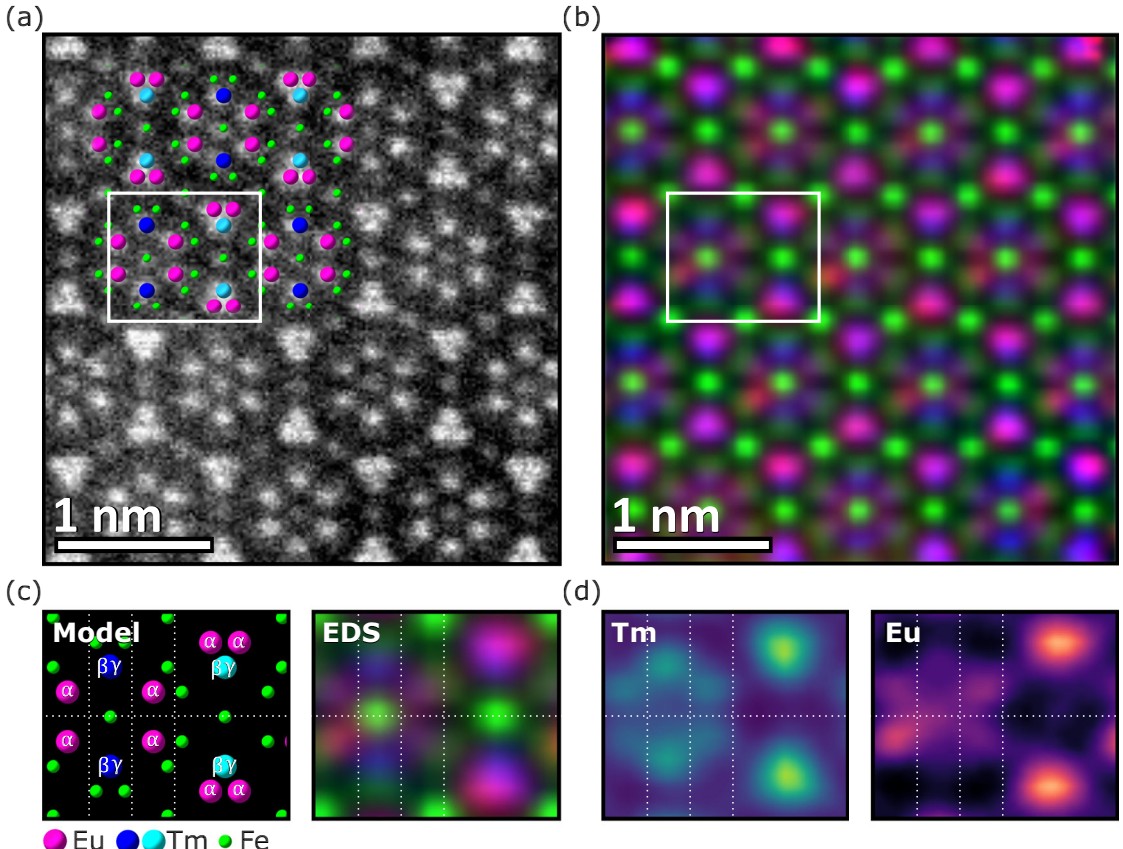

**Fig. 4 | Atomic ordering of Eu and Tm in a EuTmIG film. a** HAADF image of the [1$\bar{1}$1] zone axis in a (110) EuTmIG thin film. **b** Composite atomic resolution EDS map of the same area. **c** Model with labeled inequivalent RE sites and the equivalent region from the EDS mapping. **d** Separate EDS maps for Tm and Eu. In (**b**), the rings of six RE ions exhibit four red columns (Eu) and two blue columns (Tm). The triangles of three RE ions exhibit one blue column (Tm) and two magenta columns (mixed Tm and Eu).

This phenomenon allows the ordering to be controlled by the choice of film growth surface and rare earth composition. Moreover, we find that the magnetotaxial anisotropy derived from site ordering in EuTmIG grown by PLD is twice as large as typical values reported for garnet films grown by liquid phase epitaxy, suggesting that PLD growth leads to a stronger site selectivity.

This work resolves longstanding questions on the origin of uniaxial anisotropy in garnet films, and facilitates the development of garnet-based spintronic, photonic or magnonic devices. Furthermore, understanding the relation between site ordering and anisotropy in garnets opens a path to designing specific magnetic anisotropy landscapes via cation site occupancy, a concept that is extendable to other complex oxides and other electronic properties. The emergence of magnetotaxial anisotropy exemplifies the importance of site preference in spintronic oxides and other materials with unique site symmetries at the growth surface.

## Methods
### Growth parameters
Garnet thin films were grown by pulsed laser codeposition with a 248 nm Compex pro KrF laser at 350 mJ per pulse (flux ~ 2 mJ/cm$^2$), at a repetition rate of 10 Hz. The chamber was maintained at 150 mTorr $O_2$ during deposition (5 × 10$^{-6}$ Torr base pressure). Films were deposited on (111) and (110) GGG substrates from MTI Corporation at a substrate-target distance of 8 cm and substrate temperature of 750 ˚C. After deposition, the samples were cooled at 10˚/min in 150 mTorr oxygen environment. For codeposition from $Eu_3Fe_5O_{12}$ and $Tm_3Fe_5O_{12}$ stoichiometric targets, specific shot ratios (for total cycle of 35 shots) were selected to vary overall composition. (Tm:Eu − 0:35, 3:32, 5:30, 8:27,

9:26, 12:23, 16:19, 21:14, 27:8, 35:0). The cycles were repeated for a total of 10,000 shots (285 cycles).

For spin current injection and Hall effect sensing, Pt layers ~4 nm were deposited by d.c. magnetron sputtering at room temperature with a base pressure <2 × 10$^{-7}$ torr and a deposition rate of ~2 nm per minute.

### Thin film X-ray diffraction (XRD)
Rare earth ratio, $x$, and thickness were determined by peak position and fringe fitting of the symmetric 2θ-ω scan of the (444) reflection using a Bruker D8 Discover (X-ray source: CuK$_{\alpha 1}$, λ = 1.5406 Å) with Rigaku Globalfit fitting software. 2θ-ω scans were calibrated to match the reference value of the substrate peak. Several measurements were made to confirm that any deviation in the apparent substrate peak positions was a result of misalignment.

### Vibrating sample magnetometry (VSM)
Saturation magnetization and PMA were identified from in-plane and out-of-plane hysteresis loops acquired with a DMS 880 A VSM (calibrated with a Ni standard of similar sample dimensions).

### Spin Hall magnetoresistance (SMR) measurements
Photolithography of hall crosses by standard optical lithography was followed by Ar+ ion milling to define individual hall crosses. Hall effect measurements are taken with constant current injection at a frequency of 9973 Hz by applying a 5 V a.c. potential through a large resistor (10 kΩ) and the device in series (device resistance much less than 1 kΩ). Transverse voltage is fed back and read by the lock-in amplifier.

## General area detector diffraction system (GADDS) pole figure

(110) pole figures were acquired with a Bruker D8 Discover GADDS equipped with a $CoK_\alpha$ source, ¼ Eulerian cradle, and Vantec-2000 area detector. Pole figure data were collected with $\phi = 1°$ resolution for sufficient $\chi = 10.26°$, $35.26°$, and $60.26°$ to ensure coverage over the relevant areas of the pole sphere. Two-dimensional data were reduced with Bruker MulTex area 2 software[31].

## Scanning transmission electron microscopy (STEM)

Cross-sectional lamellae were created by standard preparation techniques on a RAITH VELION FIB-SEM. STEM imaging and energy-dispersive X-ray spectroscopy (EDS) were performed on a probe-corrected Thermo Fisher Themis Z equipped with Super-X detectors. Atomic resolution images and spectra were acquired at 200 kV with a convergence angle of 25 mrad. A beam current of 150 pA was used for imaging and spectroscopy. A series of 10 images were collected and processed using the Drift Corrected Frame Integration (DCFI) program in the Velox software to improve image accuracy and signal-to-noise ratio while minimizing the effects of sample drift during imaging. To minimize composition miscalculation due to peak overlapping, Tm M peaks, Eu M peaks, and Fe L peaks were selected for quantification. EDS maps were processed with non-linear principal component analysis and slight Gaussian blur to capture periodic information and eliminate Poisson noise.

## Density functional theory (DFT) calculations

Ab initio calculations were performed using the pseudopotential projector-augmented wave method[32] implemented in the Vienna ab initio simulation package (VASP)[33,34] with an energy cutoff of 529 eV for the plane-wave basis set. The Perdue–Burke–Ernzerhof (PBE) generalized gradient approximation (GGA) functional was used[35], along with an effected Hubbard $U$ correction of 4 eV for the localized 3d electrons of Fe ions[36,37] and 4 eV for Eu and Tm ions since the 4f electrons are treated as valence electrons[38,39] following the same methods of Nakamoto et al.[40]. In addition, for direct comparison with the work by Nakamoto et al., the revised PBE functional for solids[41] was also used in addition to the original PBE functional. A 6x6x6 Γ-centered k-point mesh was used to sample the Brillouin zone. Convergence in the self-consistent steps for the electronic structure calculation was attained once the energy difference between subsequent electronic steps was less than 1 μeV. All atomic sites in the unit cell along with the cell dimensions were relaxed using a conjugate gradient algorithm to minimize the energy with an atomic force tolerance of 0.01 eV/Å.

## Reporting summary

Further information on research design is available in the Nature Portfolio Reporting Summary linked to this article.

## Data availability

All data supporting the findings of this study are available within the manuscript and the Supplementary Information. Additional raw data may be given by the authors upon reasonable request by the reader.

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

## Acknowledgements
A.C.K. thanks the National Science Foundation Graduate Research Fellowship Program for financial support. The authors acknowledge support from NSF DMR 1808190 (C.A.R.), 2323132 (C.A.R.), and NSF ECCS-2232830 (G.S.D.B.) and the use of shared facilities of MIT.nano, CMSE, and Harvard University X-Ray Core. This work was performed in part on the Raith VELION FIB-SEM in the MIT.nano Characterization Facilities (Award: DMR 2117609). The authors also thank E. Tremsina and M. Kitcher for thoughtful discussion.

## Author contributions
Conceptualization, A.K., E.R., G.B., C.R. Methodology, A.K., E.R. Formal analysis, A.K. Investigation, A.K., E.R., Y.S., G.W., K.Y., A.P. Resources, R.G.B., G.B., C.R. Writing, Original draft, A.K. Writing, Reviewing, and Editing, A.K., E.R., G.B., C.R. Visualization, A.K. Funding acquisition, G.B., C.R. Project oversight, G.B., C.R.

## Competing interests
The authors declare no competing interests.
