## [Peer Review File · Nature Communications]

Atomic order of rare earth ions in a complex oxide: a path to magnetotaxial anisotropyREVIEWER COMMENTS

Reviewer #1 (Remarks to the Author):

The paper reports the Atomic order of rare earth ions in a complex oxide: a path to magnetotaxial anisotropy. In this paper, they quantify the resulting ordering- induced 'magnetotaxial' anisotropy as a function of Eu:Tm ratio; importantly, our pulsed laser deposited films show a large magnetotaxial anisotropy, reaching 30 kJ m⁻³ for garnets with Eu:Tm ratios close to 1, which dominates all other contributions to anisotropy and provides a useful tool in the engineering of magnetic insulators. The subject of the paper is worthy of investigation. And, the manuscript is up to the academic standard of Nature Communications. However, the manuscript needs to be revised according to the comments below before being considered for publication in Nature Communications.

1. I'm not sure if the phase of the film you synthesized is pure. Please add XRD data with complete spectra for all samples displayed in the article.
2. Please add HAADF and EDS diagrams for all samples displayed in the article.
3. As shown in the Ms diagram, please add the Hc diagram of all samples shown in the article. Based on the author's research findings, calculate the MAE, Ku, eff of all samples. And analyze the relationship between Hc and Ku, eff .
4. In the abstract, please rewrite it to highlight the author's research findings.
5. In the conclusion, please rewrite it to highlight the author's research. And explain which problems the research results can solve regarding the mechanism of magnetic materials.

6. References:

- In the part of References, please check all References. the Reference style of the References is not agreement with that of Nature Communications. Thus, all the References should be carefully checked and changed into the Reference style that is agreement with the one of Nature Communications.

Reviewer #2 (Remarks to the Author):

The authors present a study that aims to test the hypothesis that chemical ordering is responsible for long-standing observations of growth-related magnetic anisotropy in ferrimagnetic garnets. There is considerable interest in garnets from the perspective of optics, spintronics, and emerging directions in quantum materials – all of which make this area of significant scientific and technical interest. The issue addressed in this manuscript is highly challenging because the garnets can have multiple chemical components, multiple symmetry inequivalent sites, and a wide range of defects. The manuscript reports a study combining structural/chemical characterization and simulations. The manuscript puts forward an important set of results that would – if substantiated – be an important advance. I have, however, a series of specific questions that lead me to question the connection between the experimental results and the conclusion that chemical ordering is observed. There are also a few areas in which the manuscript is not sufficiently precise or quantitative. If these issues can be addressed convincingly then I think the work would be of interest for publication in Nature Communications.

My specific concerns are:

- 1) The manuscript includes two primary experimental sources of insight into the ordering of rare earth ions, neither of which is completely satisfactory. The first of these is a series of x-ray diffraction studies that exhibit the formation of x-ray reflections that the authors link to chemical ordering. The primary observation is the development of normally forbidden reflections from the {110} family of planes. These reflections are present in all of the thin film materials but somewhat stronger in the layers with an Eu-Tm mixtures. The presentation of the x-ray data, however, is highly qualitative and is missing quantitative insight. An initial concern is whether the ordering of Tm and Eu, which have relatively small difference in atomic number, could produce a reflection of sufficient intensity. A calculation comparing intensities from candidate structures with intensities from reference structures, e.g. the intensity of allowed thin film Bragg peaks, would be extremely valuable.
- 2) Again, in the consideration of the x-ray data, I am quite concerned that the authors have not ruled out other possible sources of intensity at forbidden reflections. Even the EuIG and TmIG, for example, have something like 10% of the intensity of the ordered structures. A source of concern particularly is whether defects associated with subtle differences in the growth with different compositions, some of which are invoked by the authors in their explanation of the ordering process, could lead to differences in the defect concentration and lead to intensity at forbidden reflections. I have no immediate suggestions for how to address this – but fear that this concern could greatly undermine the x-ray study.
- 3) The second major characterization component was a transmission electron microscopy study of chemical ordering. My concern in this case arises from what might just be a misinterpretation of the wording of the paper – but might also be more fundamental. Line 178 of the paper says that the team selected a “(110)-oriented” film for the TEM study. That could refer either to the growth orientation, i.e. on a (110) substrate, or to the orientation of the foil used in the TEM. If (110) refers to the growth orientation then the manuscript has a hole in its logic that needs to be addressed because the simulations and x-ray scattering studies consider (111)-oriented layers and growth processes. Does the same issue that leads to ordering in 110 oriented films lead to ordering in 111 oriented films? There are also several specific points that need to be addressed, in my opinion:
- 4) The TEM results say that the EDS shows that there is a site preference for different rare-earth elements. The results are shown in Fig. 4 as a colorful plot with no quantitative interpretation. The authors should indicate what degree of ordering is consistent with their results – or would be required to produce the observed images. Without this information the comparison with the DFT is impossible.
- 5) The thin-film diffraction patterns in Fig. 2(a) are somewhat puzzling because the GGG substrate does not appear at the same angle in each pattern. This could simply be an alignment issue – and likely does not affect the measurement of the lattice parameters. But it will be confusing to non-expert readers.
- 6) The authors should include the angular widths of the x-ray rocking curves (i.e. the mosaic width) if they have them. That would help with the logical case that defects do not contribute differently to the 110 reflections for mixed-composition films.
- 7) Line 139 mentions “measured shear strain”. Unless I missed it the manuscript does not describe what this is or how it was measured.

Reviewer #3 (Remarks to the Author):

The phenomenon of growth-induced anisotropy in garnet crystals has long been recognized. Callen's cation-ordering scenario has served as a prominent model to elucidate this phenomenon. Through the application of high-angle annular dark field imaging in STEM and

atom-resolved EDS methods, Kaczmarek et al. present compelling evidence supporting the existence of site-ordering of RE cations in the (111) growth plane of iron garnet films with mixed RE.

The formation of a three-dimensionally ordered sublattice becomes evident through atom-scale imaging. It is self-evident that such ordering breaks the symmetry associated with site occupation, resulting in an additional source of magnetic anisotropy. Despite the clarity in the experimental findings and their illustration, I have several concerns that need to be made straight:

- 1) The authors do not specify whether the observed ordered sublattice zones uniformly exist or are localized in certain areas on the sample surface.
- 2) Why did the author choose Eu and Tm rare earth elements? Whether such locally ordered structure exists in pure YIG films?
- 3) It is well-known that there are many dislocations in YIG film or single crystals (see JACOM, 966, 2023, 171527). Some Fe³⁺ ions may change to Fe²⁺ ions, meanwhile, it may cause some oxygen defects, such as oxygen vacancies and adsorbed oxygen. Whether the metastable structure is directly related to the defects?
- 4) Generally, superlattices tend to be stable upon formation. The authors describe them as metastable structures, however, it remains unclear if they provide direct evidence supporting this characterization.
- 5) Could the authors offer insights into why RE cations exhibit strong preferences for particular sites in the (111) growth plane? An exploration of the underlying factors contributing to this preference would enhance the understanding of the observed phenomena.

Responses to Reviewers:

We very much appreciate the positive reviews and the detailed comments provided by the reviewers. We have addressed all of the comments in detail, and we hope that these changes will be satisfactory.

The reviewers' comments are in red, our responses are in black and the changes to the article are summarized in blue. The edits to the article and SI are indicated with yellow highlight.

Reviewer 1

The paper reports the Atomic order of rare earth ions in a complex oxide: a path to magnetotaxial anisotropy. In this paper, they quantify the resulting ordering- induced 'magnetotaxial' anisotropy as a function of Eu:Tm ratio ; importantly, our pulsed laser deposited films show a large magnetotaxial anisotropy, reaching 30 kJ m⁻³ for garnets with Eu:Tm ratios close to 1, which dominates all other contributions to anisotropy and provides a useful tool in the engineering of magnetic insulators. The subject of the paper is worthy of investigation. And, the manuscript is up to the academic standard of Nature Communications. However, the manuscript needs to be revised according to the comments below before being considered for publication in Nature Communications.

- 1) I'm not sure if the phase of the film you synthesized is pure. Please add XRD data with complete spectra for all samples displayed in the article.

Thank you for your comment that the manuscript reaches the academic standard of the Journal.

We confirm that the films are single phase epitaxial garnets. Phase purity of these films is evident by the sole presence of peaks corresponding to the iron garnet crystal structure. We added a wide angle scan to confirm this in the SI.

Additionally, no evidence of other phases is found in direct imaging of the mixed rare-earth iron garnet films, as shown in Fig. S9. Comparison with peak positions of other possible compounds such as orthoferrites shows no peak matching.

Changes to the article:

- In SI Note 2, 2theta-omega scans from 20 – 80 degrees have been added for mixed EuTmIG films as figure S2(d) and a comment has been added in the text.

- 2) Please add HAADF and EDS diagrams for all samples displayed in the article.

The article describes six EuTmIG films with different compositions grown on (111) GGG, as well as one EuTmIG film (x = 0.5) on (110) GGG. We used only the (110) film for STEM EDS analysis of site order, because the reduced symmetry reduced the number of variants present and allows for the visualization of distinct columns of differently occupied sites by STEM.

To address the reviewer's request, we have collected STEM data on a (111) film and have included it in the SI Note 6, Figure S9. This shows the high structural quality of the film, but EDS analysis does not reveal site ordering.

(111)-oriented films cannot demonstrate site-ordering by EDS because the overlapping of the three symmetry-related ordering variants in the lamella averages out the order. For the (111)-oriented films, there is no one zone axis that can be selected to show ordered columns of atoms for all three variants.

This was explained in the main text, page 9 and derived in SI Note 5, where we comment that the nature of the order – the fact that the orientation variant can vary from cell to cell – means that cation ordering would necessarily be averaged through any column of the TEM lamella. Thus, STEM analysis cannot determine the presence or absence of cation ordering in the (111)-oriented films. The fact that we do not see ordering in the EDS of the (111) film indicates that the size of the variants is smaller than the lamella thickness of ~10 nm.

Instead, the presence of cation ordering in (111) films was confirmed by X-ray diffraction measurements (main text Fig. 3 and p7), which can detect reflections from each of the order variants. Indeed, XRD reveals all three of the site-ordered variants as seen in Fig. 3.

Changes to the article:

- The main text was modified to clarify the reasoning for doing STEM on a film with (110) orientation (page 9)
- SI Note 6 has been amended to include images of $\langle 110 \rangle$ zone axes of (111) grown EuTmIG as Figure S9.
- SI Note 6 was modified to include a discussion of limits on the size of ordered domains.

3) As shown in the Ms diagram, please add the Hc diagram of all samples shown in the article. Based on the author's research findings, calculate the MAE, Ku, eff of all samples. And analyze the relationship between Hc and Ku, eff .

The coercivity Hc, measured for an easy-axis field, depends on a number of factors including the micromagnetic processes for nucleation of reverse domains and the presence of pinning sites for domain walls. Hc is also affected by roughness of the substrate and film thickness, and it depends on the shape and sample geometry so differs between the unpatterned film and the Hall bars. Hc generally increases with the Ku (or MAE) because domain wall nucleation requires more energy when Ku is higher, but there is not a simple relationship between Hc and Ku.

MAE (or Ku) are the key parameters in the work, and we determine these from the hard-axis SMR measurements, which are based on the coherent rotation of the magnetization in a hard-axis field. Thus, the easy-axis domain nucleation and motion processes are not related to the measurement. We therefore did not focus specifically on discussing Hc, but we have added a table to SI Note 3 to report the values.

Changes to the article:

- A new table S4 and comment have been added to SI Note 3 with the coercive fields (H_c) of each of the unpatterned films (subsequent tables were renumbered.)
- 4) In the abstract, please rewrite it to highlight the author's research findings.
 - 5) In the conclusion, please rewrite it to highlight the author's research. And explain which problems the research results can solve regarding the mechanism of magnetic materials.

Changes to the article:

- The abstract and conclusion have been revised to highlight the methods and findings of a unique growth-induced anisotropy in mixed rare earth iron garnet films. The author has emphasized that she has revealed the atomic origins of this anisotropy, indicates the implications of the work and explains that understanding of this phenomenon can be used to help engineer materials with desirable properties for spintronic and magnonic devices.
- 6) References:
 - In the part of References, please check all References. the Reference style of the References is not agreement with that of Nature Communications. Thus, all the References should be carefully checked and changed into the Reference style that is agreement with the one of Nature Communications.

Changes to the article: References have been checked and missing data added.

Reviewer 2

The authors present a study that aims to test the hypothesis that chemical ordering is responsible for long-standing observations of growth-related magnetic anisotropy in ferrimagnetic garnets. There is considerable interest in garnets from the perspective of optics, spintronics, and emerging directions in quantum materials – all of which make this area of significant scientific and technical interest. The issue addressed in this manuscript is highly challenging because the garnets can have multiple chemical components, multiple symmetry inequivalent sites, and a wide range of defects. The manuscript reports a study combining structural/chemical characterization and simulations. The manuscript puts forward an important set of results that would – if substantiated – be an important advance. I have, however, a series of specific questions that lead me to question the connection between the experimental results and the conclusion that chemical ordering is observed. There are also a few areas in which the manuscript is not sufficiently precise or quantitative. If these issues can be addressed convincingly then I think the work would be of interest for publication in Nature Communications.

My specific concerns are:

- 1) The manuscript includes two primary experimental sources of insight into the ordering of rare earth ions, neither of which is completely satisfactory. The first of these is a series of x-ray diffraction studies that exhibit the formation of x-ray reflections that the authors link to chemical ordering. The primary observation is the development of normally forbidden reflections from the {110} family of planes. These reflections are present in all of the thin film materials but somewhat stronger in the layers with an Eu-Tm mixtures. The presentation of the x-ray data, however, is highly qualitative and is missing quantitative insight. An initial concern is whether the ordering of Tm and Eu, which have relatively small difference in atomic number, could produce a reflection of sufficient intensity. A calculation comparing intensities from candidate structures with intensities from reference structures, e.g. the intensity of allowed thin film Bragg peaks, would be extremely valuable.
- 2) Again, in the consideration of the x-ray data, I am quite concerned that the authors have not ruled out other possible sources of intensity at forbidden reflections. Even the EuIG and TmIG, for example, have something like 10% of the intensity of the ordered structures. A source of concern particularly is whether defects associated with subtle differences in the growth with different compositions, some of which are invoked by the authors in their explanation of the ordering process, could lead to differences in the defect concentration and lead to intensity at forbidden reflections. I have no immediate suggestions for how to address this – but fear that this concern could greatly undermine the x-ray study.

Thank you for your overall comment that the article addresses an area of significant scientific and technical interest and the work would be of interest for the Journal if the comments are addressed. We start by answering the first two related comments from a modeling perspective. We carried out the following simulations of XRD to demonstrate the evolution of the ‘forbidden’ peak:

- (a) We simulated the powder diffraction corresponding to GGG (a common substrate, providing a reference), in addition to patterns for the Tm,Eu garnets corresponding to the site-ordering expected for (111), (110), and (112) growth surfaces.
- (b) We compared the effect of perfect order and partial order 50:50 mix of Eu:Tm for site order corresponding to the [111] growth direction.

The peak intensity depends on the structure factor which varies with the difference in atomic number of the ordered RE cations. To address the specific question of the reviewer, our simulations show that Eu ($Z=63$) and Tm ($Z=69$) are sufficiently different in atomic number to produce a measurable peak intensity in the ordered state.

The results are described in a new Supplemental Note 8. Briefly, they show that the peak intensity is highest for the best ordered structure (as expected) and decreases quadratically with degree of order. These simulations also show that as the symmetry of the growth surface

decreases (i.e. comparing (111), (110), and (112) growth surfaces), more ‘forbidden’ peaks emerge. We note that these simulations represent the powder diffraction pattern, and the epitaxial nature of the film and the presence of variants determine which peaks are accessible experimentally.

We are very interested in quantifying the degree of site ordering in these materials. The conventional XRD used here can be difficult to compare quantitatively between samples, but the data is clear that the 50:50 Eu:Tm has several times greater intensity of the ‘forbidden’ peak than the end members. Ideally the end members will have zero peak, like the GGG substrate, but stoichiometry deviations (oxygen vacancies or cation antisites) could produce peak intensity. However, it is reasonable to assume that the population of such point defects will be similar for all the films, due to their same growth parameters, so the much greater forbidden peak intensity for the mixed-RE films vs. the end members is a consequence of the ordering.

Changes to the article:

- A supplemental note 8 has been added to explain using simulations the emergence of the forbidden peak due to site preference of Eu and Tm (including maximum expected intensity).
- Comments have been added in the main text, page 8.

3) The second major characterization component was a transmission electron microscopy study of chemical ordering. My concern in this case arises from what might just be a misinterpretation of the wording of the paper – but might also be more fundamental. Line 178 of the paper says that the team selected a “(110)-oriented” film for the TEM study. That could refer either to the growth orientation, i.e. on a (110) substrate, or to the orientation of the foil used in the TEM. If (110) refers to the growth orientation then the manuscript has a hole in its logic that needs to be addressed because the simulations and x-ray scattering studies consider (111)-oriented layers and growth processes. Does the same issue that leads to ordering in 110 oriented films lead to ordering in 111 oriented films?

The reviewer is correct in understanding that the TEM was carried out on a (110)-oriented film (i.e. substrate normal is [110]) whereas the x-ray studies and other data are measured on (111)-oriented films. However, referring to the last sentence of the reviewer’s comment, both (111) and (110)-oriented films are expected to exhibit RE site ordering. Because of the different sets of orientations of the RE sites within a unit cell, the RE sites are inequivalent for *any* growth direction. Hence one expects site ordering and the resulting magnetotaxial anisotropy for both (110) and (111) (and indeed for all other growth directions), as pointed out by Callen in his 1971 theoretical treatment. The magnitude of the anisotropy for different growth directions is not expected to be the same, but site ordering is expected to be present in both. SI Note 1 describes the inequivalent sites for several growth directions.

The reason for doing TEM on the (110)-oriented film is because of the lower number of variants of the ordered structure. For (111) films there are three possible variants of the ordered unit cell,

which will be superposed in the TEM making it impossible to identify the presence of ordering from the EDS map. In contrast, for the (110) oriented sample imaged along the in-plane [-110] zone axis, there is only one variant and the site ordering is detectable.

We clarified these points in the text. We also added XRD simulations for the (110) and (112) growth directions to the new Supplementary Note 8 to show that even though the order will be different for different growth orientations, it will still lead to the emergence of an order peak.

Changes to the article:

- In SI Note 8 we added simulations of the emergence of forbidden peaks in (110) and other orientations to show that x-ray diffraction can be used to detect site ordering in films of various orientation.
- We clarified the reason for using (110) oriented film for TEM, page 9.

There are also several specific points that need to be addressed, in my opinion:

- 1) The TEM results say that the EDS shows that there is a site preference for different rare-earth elements. The results are shown in Fig. 4 as a colorful plot with no quantitative interpretation. The authors should indicate what degree of ordering is consistent with their results – or would be required to produce the observed images. Without this information the comparison with the DFT is impossible.

The EDS map in Fig. 4 is semiquantitative because different x-ray peaks need to be selected for the Tm, Eu and Fe elements due to overlap. The procedure for generating the Fig. 4 image is described in Supplementary Note 6, and includes a non-linear principal component analysis and Gaussian filtering. The unprocessed data was already given in Fig. S12 and still shows the intensity differences corresponding to site ordering, but with more noise than Fig. 4.

To help address the reviewer's question, we extracted line scans to show the relative amounts of Eu and Tm in each type of site. This indicates the presence of site ordering in a more quantitative manner.

Changes to the article:

- Line scans added in Supplementary Fig. S13, and comment added in Note 6 to describe the result.
- 2) The thin-film diffraction patterns in Fig. 2(a) are somewhat puzzling because the GGG substrate does not appear at the same angle in each pattern. This could simply be an alignment issue – and likely does not affect the measurement of the lattice parameters. But it will be confusing to non-expert readers.

The variation of the position of the GGG peak is due to an alignment issue with the instrument. To confirm this we remeasured some of the samples with additional attention to alignment to show that the GGG peak does match the bulk value. The misalignment did not affect the later

calculations of strain. To avoid ambiguity, we replotted the diffraction patterns by normalizing the substrate peak to the literature value for GGG.

Changes to the article:

- Figure 2(a) was replotted by correcting for alignment.
- 3) The authors should include the angular widths of the x-ray rocking curves (i.e. the mosaic width) if they have them. That would help with the logical case that defects do not contribute differently to the 110 reflections for mixed-composition films.

Changes to the article:

- In SI Note 2, a column in table S3 has been added with the full width-half max values of the rocking curve peaks for each of the films. All values show very little mosaic spread.
- 4) Line 139 mentions “measured shear strain”. Unless I missed it the manuscript does not describe what this is or how it was measured.

A description of how strain is calculated from XRD is found in SI Note 2.

Changes to the article:

- A reference to SI Note 2 was added on p6.

Reviewer 3

The phenomenon of growth-induced anisotropy in garnet crystals has long been recognized. Callen's cation-ordering scenario has served as a prominent model to elucidate this phenomenon. Through the application of high-angle annular dark field imaging in STEM and atom-resolved EDS methods, Kaczmarek et al. present compelling evidence supporting the existence of site-ordering of RE cations in the (111) growth plane of iron garnet films with mixed RE. The formation of a three-dimensionally ordered sublattice becomes evident through atom-scale imaging. It is self-evident that such ordering breaks the symmetry associated with site occupation, resulting in an additional source of magnetic anisotropy. Despite the clarity in the experimental findings and their illustration, I have several concerns that need to be made straight:

- 1) The authors do not specify whether the observed ordered sublattice zones uniformly exist or are localized in certain areas on the sample surface.

Thank you very much for your comments recognising the compelling evidence presented in the article.

In Fig. 3, three order peaks are present, corresponding to the three variants of the site-ordered structure, but the scan averages over a mm-size area so does not reveal the size of the variants. From symmetry arguments we expect the three variants to be present in equal volumes, and this is consistent with PMA (rather than a tilted anisotropy expected if only one variant were present). The authors believe that “domains” of these variants are very small, on the scale of a few unit cells or less, due to the evidence from the STEM of the (111)-oriented film. The lamella was ~10 nm thick, but the order was averaged over that thickness. This indicates that the size of the oriented regions is smaller than 10 nm.

Changes to the article:

- SI Note 6 was modified to include a discussion of limits on the size of ordered domains.
- 2) **Why did the author choose Eu and Tm rare earth elements? Whether such locally ordered structure exists in pure YIG films?**

Eu and Tm are chosen as the rare earth elements due to large ionic radii difference, which promotes a strong tendency for site ordering and hence a larger magnetotaxial anisotropy [Callen, *Materials Research Bulletin* **6**, 931–938 (1971)]. Furthermore, it was also convenient that EuIG and TmIG have opposite strain states when grown on GGG, and they have opposite signs of magnetostriction. This means that while the end-members have PMA due to magnetostriction, intermediate Eu:Tm compositions have low or zero strain, and low or zero magnetostriction. Thus, the idea of creating PMA without reliance on strain can be tested. Growth-induced anisotropy occurs in many different mixed RE garnets [Eschenfelder, *Magnetic Bubble Technology*. (Springer-Verlag, 1980)]. However, it is not expected in stoichiometric non-defective YIG films because all the dodecahedral sites are filled with the same ion (Y^{3+}).

The choice of Eu:Tm was discussed in the main text, page 4.

- 3) **It is well-known that there are many dislocations in YIG film or single crystals (see JACom, 966, 2023, 171527). Some Fe³⁺ ions may change to Fe²⁺ ions, meanwhile, it may cause some oxygen defects, such as oxygen vacancies and adsorbed oxygen. Whether the metastable structure is directly related to the defects?**

Dislocations can occur in epitaxial garnet films as noted by the reviewer, but our observations of both the (110) and the (111)-oriented EuTmIG (Fig. 1 and Supplementary Note 6), show no dislocations across the entire visible lamella. This is consistent with many other TEM investigations of epitaxial RE, Bi or Y garnets in our prior work, e.g. *Nat Commun* **11**, 1090 (2020), *Small* **19** 2300824 (2023), etc. Furthermore, the strain in the films does not relax even for thicknesses of 10s of nm, according to RSM data, suggesting that dislocations do not form. Hence we do not believe dislocations play a major role in the site occupation or growth-induced anisotropy.

(As a contrast, we have observed dislocations in garnets codeposited with BaTiO₃ where the substituents cause significant lattice distortion (Appl. Phys. Lett. 121, 231604 (2022)).)

Regarding Fe²⁺, we performed X-ray absorption spectroscopy on TbIG films in a prior work (*Small* **19** 2300824 (2023)) to quantify the Fe valence state and site occupancy. The amount of Fe²⁺ was less than about 3%. Therefore we do not believe it plays an important role in the RE site ordering. Further, we would expect all the films to show similar amounts of Fe²⁺ or other point defects such as vacancies because they were grown under similar conditions, so the dramatic differences in anisotropy between end-members and the mixed RE garnet should be attributed to the RE ordering and not to Fe²⁺ or vacancies.

Changes to the article:

- A paragraph on the role of dislocations and point defects was added to Supplementary Note 6.
- 4) **Generally, superlattices tend to be stable upon formation. The authors describe them as metastable structures, however, it remains unclear if they provide direct evidence supporting this characterization.**

We emphasize that these site-ordered structures are not conventional superlattices with layers, but they instead represent a 3D ordering based on the geometrical orientation of the 24 dodecahedral sites in the unit cell. The ordered structures are formed during the growth of these thin films by the preference of arriving RE ions to order between inequivalent dodecahedral sites. Once formed, the ordering is highly stable at ambient conditions, and prior work has shown that the growth-induced anisotropy is robust up to high temperatures (e.g. Eschenfelder, Magnetic Bubble Technology, 1980) which indicates that the growth-induced order is retained to high temperatures. The order and the resulting anisotropy can be lost by a sufficiently high temperature anneal (e.g. >600°C) which allows for RE diffusion within the structure. We have observed this in our films but have not fully analysed the kinetics.

The site-ordered structures are not equilibrium phases like e.g. a FePt alloy which can have ordered (L1₀) or disordered (fcc) structures at different temperatures, or a double perovskite with a stable ordered structure. This is clear because (1) the type of order depends on the growth orientation, and (2) the order is formed in epitaxial thin films but not in bulk crystals of mixed-RE garnets. One can consider the ordered structure as a metastable arrangement that forms during growth and is kinetically trapped at temperatures up to several hundred °C. This was noted in the main text p4.

Changes to the article:

- A paragraph was added to Supplementary Note 1 to discuss the stability.

- 5) Could the authors offer insights into why RE cations exhibit strong preferences for particular sites in the (111) growth plane? An exploration of the underlying factors contributing to this preference would enhance the understanding of the observed phenomena.

The reason for site preference was hypothesized by Callen and is summarized in SI Note 1. It occurs because the dodecahedral sites for the REs, presented at the surface of the film, are differently oriented with respect to the growth direction and therefore have a different geometry to capture arriving RE adatoms. REs of different ionic sizes therefore segregate between these inequivalent sites. Hence the tendency to site order depends on the ionic radius difference, as found empirically (Eschenfelder, Magnetic Bubble Technology, 1980).

Changes to the article:

- Comments on the origin of the site preference are found in the paragraph that we added to Supplementary Note 1.

REVIEWER COMMENTS

Reviewer #1 (Remarks to the Author):

According to the reviewers' comments, the authors have carefully revised the paper "NCOMMS-23-57326A". The paper reports the Atomic order of rare earth ions in a complex oxide: a path to magnetotaxial anisotropy. The subject of the paper is worthy of investigation. And the revised manuscript is up to the academic standard of Nature Communications. Thus, the manuscript could be considered for publication in Nature Communications.

Reviewer #2 (Remarks to the Author):

With the exception of a few remaining points below, the authors have addressed my concerns adequately.

1) My original review expressed the concern that there are multiple possible sources of intensity at the locations of nominally forbidden reflections besides the ordering of cations. There is clear evidence for this in the authors work: the intensities of the end-member compounds are not zero. The authors say in their response: "... the data is clear that the 50:50 Eu:Tm has several times greater intensity of the 'forbidden' peak than the end members. Ideally the end members will have zero peak, like the GGG substrate, but stoichiometry deviations (oxygen vacancies or cation antisites) could produce peak intensity. However, it is reasonable to assume that the population of such point defects will be similar for all the films, due to their same growth parameters, so the much greater forbidden peak intensity for the mixed-RE films vs. the end members is a consequence of the ordering." The discussion of this critical assumption is missing, as far as I can tell, from the revised manuscript. Without it the x-ray data has the potential to be misinterpreted as more authoritative than it is.

2) I expressed the concern that the TEM and x-ray data considered different orientations of the sample. This has been addressed in large part, but raises the question of whether an x-ray study of the forbidden reflections of the (110)-oriented layer is available. If so, it would be a useful addition to the manuscript.

3) The authors have replotted Fig. 2b to take care of the misalignment issue. A point should be made in the methods or supplementary information to indicate that this was done.

Reviewer #3 (Remarks to the Author):

The revised paper and the responses addressed all my concerns, I will recommend acceptance of this paper.

Responses to Second Review:

We very much appreciate the positive reviews and the detailed comments provided by the reviewers. We have addressed all of the comments of Reviewer 2 in detail, and we hope that these changes will be satisfactory.

The reviewers' comments are in black, our responses are in red and the edits to the article and SI are indicated with yellow highlight.

REVIEWER COMMENTS – Second Review

Reviewer #1 (Remarks to the Author):

According to the reviewers' comments, the authours have carefully revised the paper "NCOMMS-23-57326A". The paper reports the Atomic order of rare earth ions in a complex oxide: a path to magnetotaxial anisotropy. The subject of the paper is worthy of investigation. And the revised manuscript is up to the academic standard of Nature Communications. Thus, the manuscript could be considered for publication in Nature Communications.

Thank you for your comments.

Reviewer #2 (Remarks to the Author):

With the exception of a few remaining points below, the authors have addressed my concerns adequately.

- 1) My original review expressed the concern that there are multiple possible sources of intensity at the locations of nominally forbidden reflections besides the ordering of cations. There is clear evidence for this in the authors work: the intensities of the end-member compounds are not zero. The authors say in their response: "... the data is clear that the 50:50 Eu:Tm has several times greater intensity of the 'forbidden' peak than the end members. Ideally the end members will have zero peak, like the GGG substrate, but stoichiometry deviations (oxygen vacancies or cation antisites) could produce peak intensity. However, it is reasonable to assume that the population of such point defects will be similar for all the films, due to their same growth parameters, so the much greater forbidden peak intensity for the mixed-RE films vs. the end members is a consequence of the ordering." The discussion of this critical assumption is missing, as far as I can tell, from the revised manuscript. Without it the x-ray data has the potential to be misinterpreted as more authoritative than it is.
- 2) I expressed the concern that the TEM and x-ray data considered different orientations of the sample. This has been addressed in large part, but raises the question of whether an x-ray study of the forbidden reflections of the (110)-oriented layer is available. If so, it would be a useful addition to the manuscript.
- 3) The authors have replotted Fig. 2b to take care of the misalignment issue. A point should be made in the methods or supplementary information to indicate that this was done.

We appreciate the additional comments, and we have addressed them in the second revision.

- 1) The reviewer notes that the forbidden intensities of the end-member TmIG and EuIG (110) peaks are not zero, and that we responded to this in our first revision by commenting that the endmember forbidden peaks have several times lower intensity than that of the 50:50 EuTmIG which supports the assertion that the forbidden peak in the 50:50 EuTmIG comes mainly from Eu/Tm site ordering. The reviewer asks for additional support of this comment. To address this, we carried out simulations of the effect of a common point defect (an oxygen vacancy) on the forbidden reflections in garnet. We showed that even one vacancy in a unit cell of GGG, corresponding to a stoichiometry of $Gd_3Fe_5O_{11.875}$, would lead to a non-zero (110) intensity. Several authors including ourselves have shown that vacancies are common in garnet thin films grown by PLD (Rosenberg, E. *et al.* Revealing Site Occupancy in a Complex Oxide: Terbium Iron Garnet. *Small* 2300824 2023). Therefore, it is plausible that the weak (110) forbidden peaks in the asymmetric scans of endmember films come from such point defects. We added a detailed discussion of this in the Supplementary Information Note 8 (page 30) and provided supporting data in Figure S16. This is referenced in the main text p8.
- 2) The reviewer asked for an x-ray study of the (110)-oriented film, to ensure that the x-ray data from the (111) films can be related to the TEM data from (110) films. We carried out a study of the forbidden reflections in the (110) film of the 50:50 EuTmIG and EuIG films. This is described in a new Supplementary Note 9. We were able to measure 'forbidden' (110) and (330) reflections of the films. The (110) intensity in the bare GGG as well as the films is explained by 'umweganregung' in the symmetric XRD geometry, but the (330) peak of the EuTmIG film is clearly more intense than that of the EuIG and the GGG (which has no peak) and corresponds to the correct d-spacing for the film, not GGG. This supports the presence of site ordered RE in the EuTmIG. This is referenced in the main text p9.
- 3) We added a comment in the Methods section of the Supplementary Information to address the calibration issue in Figure 2b. (We actually remeasured the data for several samples to confirm that any apparent peak shift was a result of misalignment.)

Reviewer #3 (Remarks to the Author):

The revised paper and the responses addressed all my concerns, I will recommend acceptance of this paper.

Thank you for your comments.

REVIEWERS' COMMENTS

Reviewer #2 (Remarks to the Author):

The authors have addressed my concerns very thoughtfully and comprehensively. My opinion is that the manuscript is now suitable for publication and that the authors should be complemented on their findings and analysis.

Responses to Third Review:

We greatly appreciate the thorough critique from each of the reviewers, as we think it has improved the strength of our findings and fitness for the journal.

The reviewers' comments are in black, our responses are in red.

REVIEWERS' COMMENTS – Third Review

Reviewer #2 (Remarks to the Author):

The authors have addressed my concerns very thoughtfully and comprehensively. My opinion is that the manuscript is now suitable for publication and that the authors should be complemented on their findings and analysis.

Thank you for your feedback and detailed review.